# The Trithorax protein Ash1L promotes myoblast fusion by activating Cdon expression

Ilaria Castiglioni[1], Roberta Caccia [1], Jose Manuel Garcia-Manteiga[2], Giulia Ferri[1], Giuseppina Caretti [3], Ivan Molineris[2], Kenichi Nishioka [4,5] & Davide Gabellini [1]

Myoblast fusion (MF) is required for muscle growth and repair, and its alteration contributes to muscle diseases. The mechanisms governing this process are incompletely understood, and no epigenetic regulator has been previously described. Ash1L is an epigenetic activator belonging to the Trithorax group of proteins and is involved in FSHD muscular dystrophy, autism and cancer. Its physiological role in skeletal muscle is unknown. Here we report that Ash1L expression is positively correlated with MF and reduced in Duchenne muscular dystrophy. In vivo, ex vivo and in vitro experiments support a selective and evolutionary conserved requirement for Ash1L in MF. RNA- and ChIP-sequencing indicate that Ash1L is required to counteract Polycomb repressive activity to allow activation of selected myogenesis genes, in particular the key MF gene *Cdon*. Our results promote Ash1L as an important epigenetic regulator of MF and suggest that its activity could be targeted to improve cell therapy for muscle diseases.

---

[1] Division of Genetics and Cell Biology, IRCCS San Raffaele Scientific Institute, via Olgettina 60, Milano 20132, Italy. [2] Center for Translational Genomics and BioInformatics, IRCCS San Raffaele Scientific Institute, via Olgettina 60, Milano 20132, Italy. [3] Department of Biosciences, University of Milan, via Celoria 26, Milano 20133, Italy. [4] Department of Biomolecular Sciences, Division of Molecular Genetics and Epigenetics, Faculty of Medicine, Saga University, Saga, Japan. [5] Laboratory for Developmental Genetics, RIKEN IMS, 1-7-22 Suehiro-cho, Tsurumi-ku, Yokohama City, Kanagawa 230-0045, Japan. Correspondence and requests for materials should be addressed to D.G. (email: gabellini.davide@hsr.it)

Cell fusion is a fundamental process required for sexual reproduction and for the development of many body tissues[1,2]. During fertilization, the fusion of sperm and egg leads to the formation of a diploid cell[3,4]. Macrophages undergo fusion events to form osteoclasts during bone development[5]. Also, the genetic reprogramming of somatic cells could occur following stem cell fusion[6]. Besides these important examples, a specific and tightly regulated type of fusion is crucial for skeletal muscle tissue formation, growth, and repair[1,7,8]. During embryonic development, mononucleated muscle cells, termed myoblasts, undergo massive proliferation, providing the required

number of precursor cells to build skeletal muscles. Subsequently, they exit the cell cycle, start to differentiate and fuse with one another to generate multinucleated and fully differentiated myofibers[7,8]. This process continues for the first 3 weeks of postnatal life, in mice[9]. In the adulthood at steady state, myoblast fusion events are sporadic but necessary for muscle regeneration. In response to muscle damage, muscle stem (satellite) cells, which normally reside at the periphery of the mature fiber in a quiescent state, activate, proliferate, and fuse to reconstruct the damaged muscle[10–14]. Notably, defects in myoblast fusion contribute to the pathogenesis of an increasing number of muscle diseases[15–20].

Recognition and adhesion between fusion partners is an early and key event in myoblast fusion[21]. Accordingly, the majority of the proteins involved in myoblast fusion identified so far are adhesion proteins, transmembrane lipids, intracellular signaling, or adaptor proteins[22–25]. Among these, the immunoglobulin-like and fibronectin type III domains protein Cdon (cell-adhesion-molecule-related/downregulated by oncogenes) is a member of a cell-surface receptor complex that mediates cell–cell interactions between muscle precursor cells and positively regulates myogenesis[26,27].

While we know extensively about the structural and signaling components required for myoblast fusion, the gene expression regulation underlying this important process is poorly understood. A few transcription factors activating myoblast fusion have been described[28–31]; however, an identification of fusion-regulated target genes on a genome-wide scale is mostly lacking[28–31]. Importantly, no epigenetic regulator of myoblast fusion has ever been defined.

Absent, small, or homeotic discs like 1 (Ash1L) is a Trithorax group protein that positively regulates gene transcription by counteracting Polycomb group (PcG) protein-mediated silencing[32]. Ash1L di-methylates histone H3 lysine 36 to produce H3K36me2[33], a histone mark associated with transcriptional activation and enriched at transcription start sites (TSS)[34]. Ash1L is involved in several pathological conditions, including autoimmune disease, autism, and cancer[34–40]. Despite its involvement in facioscapulohumeral (FSHD) muscular dystrophy[41], the physiological role of Ash1L in skeletal muscle is unknown.

We found an evolutionarily conserved requirement for Ash1L in myoblast fusion. Combining genome-wide approaches with gain-of-function and loss-of-function assays, we identified Cdon as a direct Ash1L target required for Ash1L-mediated myoblast fusion activation. Altogether, our results promote Ash1L as a crucial epigenetic regulator of myoblast fusion.

## Results

### Ash1L expression positively correlates with myoblast fusion.
To begin investigating the physiological role of Ash1L in the skeletal muscle, we evaluated its expression in three crucial processes: muscle development, muscle regeneration, and in vitro muscle differentiation (Fig. 1). During murine prenatal development and adulthood, Ash1L expression resulted maximal in fetal skeletal muscles, when myoblast fusion events are most frequent[11], and was progressively and significantly reduced reaching a minimum at P28, when myoblast fusion is normally off (Fig. 1a). Intriguingly, the key myoblast fusion factor Myomaker displayed a similar expression pattern (Fig. 1a)[22,42]. On the contrary, the gene encoding for the adult skeletal muscle myosin Myh4 showed an opposite trend, reaching a maximum when myoblast fusion is over

(Fig. 1a). In adulthood at steady state, myoblast fusion is nearly absent, but it is reactivated during regeneration in response to muscle damage[43]. To assess Ash1L expression during muscle regeneration, we analyzed tibialis anterior muscle of 8-week-old mice after cardiotoxin (CTX) injury (Fig. 1b). Compared to uninjured muscle, Ash1L expression was significantly upregulated during the initial phase of muscle regeneration (day 5), and downregulated at day 10, when myoblast fusion decreases[43] similarly to Myomaker[44]. During in vitro myogenesis, we found that Ash1L upregulation occurred just after the induction of differentiation (day 1) (Fig. 1c), and simultaneously with myoblast fusion activation[45]. A key step for myoblast fusion activation is cell–cell contact[46–48]. Interestingly, we found that cell confluence in proliferation media was sufficient to significantly induce Ash1L expression (Fig. 1d). Myoblast fusion defects have been described in several muscle diseases[15–17,49–55] including Duchenne muscular dystrophy (DMD)[56,57]. Inspection of the GEO database (http://www.ncbi.nlm.nih.gov/geo/info/profiles.html) revealed that Ash1L expression is significantly downregulated in muscle tissue from both DMD patients and the DMD mouse model mdx, which we confirmed by real-time quantitative reverse transcription PCR (RT-qPCR) (Supplementary Figure 1). Collectively, our results indicate that the expression of Ash1L is positively correlated to myoblast fusion and is significantly downregulated in DMD.

### Ash1L is specifically required for myoblast fusion.
We used embryonic stem cells containing a gene trap cassette inserted in intron 1 of the mouse Ash1L gene to generate mice lacking Ash1L (Ash1L GT). Unlike a previous report showing that the majority of Ash1L GT mice survived into adulthood, although not at a Mendelian ratio[58], most of our independently generated Ash1L GT mice were not born alive and the remaining animals displayed full lethality by P8 (Supplementary Figure 2), possibly due to the different procedures utilized and the degree of backcrossing of the different strains (see Methods). We hence decided to analyze Ash1L GT mice at embryonic day 18.5 when Ash1L expression and myoblast fusion are high. While we found no significant alteration in the number of muscle fibers, Gomori-trichrome staining of quadricep transverse cryosections revealed a significantly reduced myofiber cross-sectional area (CSA) in Ash1L GT mice compared to wild-type mice (Fig. 2a). Interestingly, longitudinal sections showed that the number of myonuclei per individual myofiber is significantly lower in Ash1L GT mice compared to wild-type mice (Fig. 2b).

To elucidate the muscle formation defect of Ash1L GT mice, we compared ex vivo satellite cell-derived myoblast cultures from hindlimbs of wild-type and Ash1L GT mice isolated at the embryonic stage of 18.5. While the percentage of myosin heavy chain (Mhc)-positive cells displayed no significant alteration, the fusion index of Ash1L GT myoblasts was significantly lower

**Fig. 1** Correlation between Ash1L expression and myoblast fusion. **a** Ash1L expression during muscle development. RT-qPCR analysis on muscle tissue from hindlimbs of mice from the embryonic stage E16.5 to adulthood (p28). Expression analysis of Myomaker, as a positive fusion gene, and adult myosin (Myh4) as a positive control for adult stages. Unpaired two-tailed t test. Confidence intervals 95%. n = 6. **b** Ash1L expression in regenerating muscle tissue. RT-qPCR analysis of Ash1L expression in tibialis anterior of wild-type adult mice, untreated (UNT), or 5 and 10 days after cardiotoxin (CTX) injection (left panel). Immunofluorescence for Ash1L (in green) and nuclear staining (Hoechst), in transverse cryosections from the tibialis anterior muscles of injured wild-type mice, 5 days after cardiotoxin injection (CTX 5 days) compared to untreated controls (Unt). Scale bar, 50 μm. Magnification ×65. Arrows indicate the Ash1L-positive nuclei. Unpaired two-tailed t test. Confidence intervals 95%. n = 4. **c** Ash1L protein level during in vitro muscle differentiation. Immunoblot of C2C12 cells at days 0 and 1, and densitometric analysis of Ash1L signal relative to vinculin as housekeeping protein (left panel). Immunofluorescence for Ash1L (in green) and nuclear staining (Hoechst), in C2C12 cells at days 0 and 1 of in vitro muscle differentiation. Scale bar, 100 μm. Magnification ×20. Paired two-tailed t test. Confidence intervals 95%. Data are the mean for three independent experiments. **d** Ash1L protein level in proliferating myoblasts vs. confluent cells. Comparison between proliferating myoblasts (P) and confluent cells (C). Paired two-tailed t test. Confidence intervals 95%. Data are the mean for three independent experiments. Source data are provided as a Source Data file. *$p \leq 0.05$, **$p \leq 0.01$, ***$p \leq 0.001$, ****$p \leq 0.0001$. NS not significant

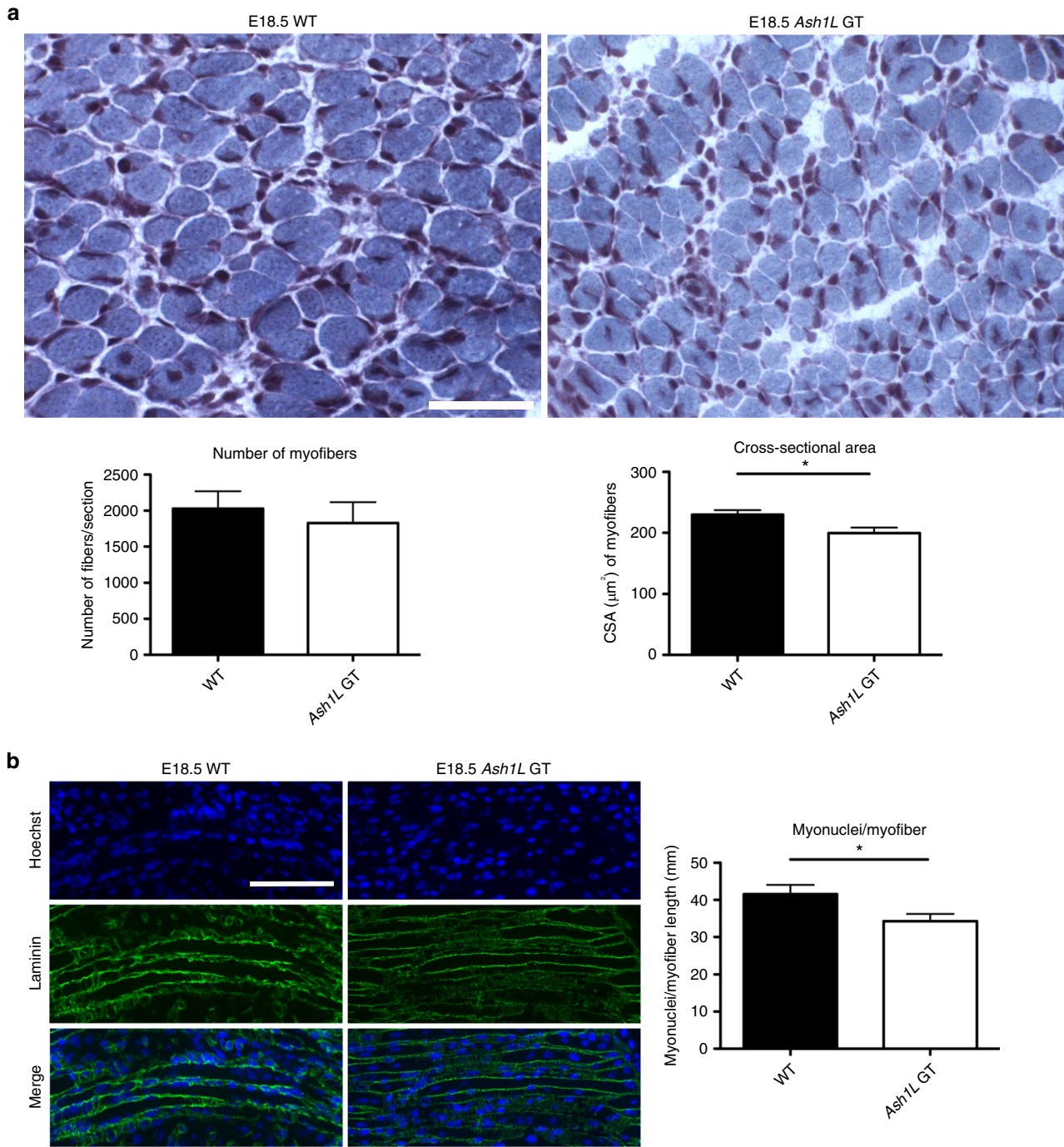

**Fig. 2** *Ash1L* GT mice display muscle hypoplasia. **a** Gomori-trichrome staining of transverse cryosections from wild-type (WT) and *Ash1L* GT mice, at embryonic stage E18.5 (upper panel). Quantification of: the total number of myofibers and myofibers cross-sectional area (CSA) (lower panel). $n = 3$. Scale bar, 100 µm, magnification ×20. Unpaired two-tailed *t* test. Confidence intervals 95%. Results come from six biological replicates. **b** Immunofluorescence for laminin (in green) and nuclear staining (Hoechst) in longitudinal cryosections from wild-type (WT) and *Ash1L* GT mice, at the embryonic stage of E18.5 (left panel). Quantification of the number of myonuclei normalized on myofiber length in mm (myonuclei/myofibers) (right panel). Scale bar, 100 µm, magnification ×20. Unpaired two-tailed *t* test. Confidence intervals 95%. Results come from seven biological replicates. Source data are provided as a Source Data file. *$p \leq 0.05$

compared to controls (Fig. 3a). Moreover, individual *Ash1L* GT myofibers presented a significantly reduced number of myonuclei compared to controls (Fig. 3a).

To further support our findings, we performed acute *Ash1L* knockdown (KD) in C2C12 muscle cells. Similar to *Ash1L* GT, *Ash1L* KD caused no significant alteration in the percentage of Mhc-positive myofibers, but a significant reduction of the fusion

index and the number of nuclei per myofiber (Fig. 3b). Importantly, similar results were obtained in human primary myoblasts (Fig. 3c), indicating that Ash1L is evolutionary required for myotube formation.

Muscle formation requires myoblast proliferation, migration, differentiation, and fusion[59]. In principle, dysfunction of one or more of these processes could contribute to aberrant myotube

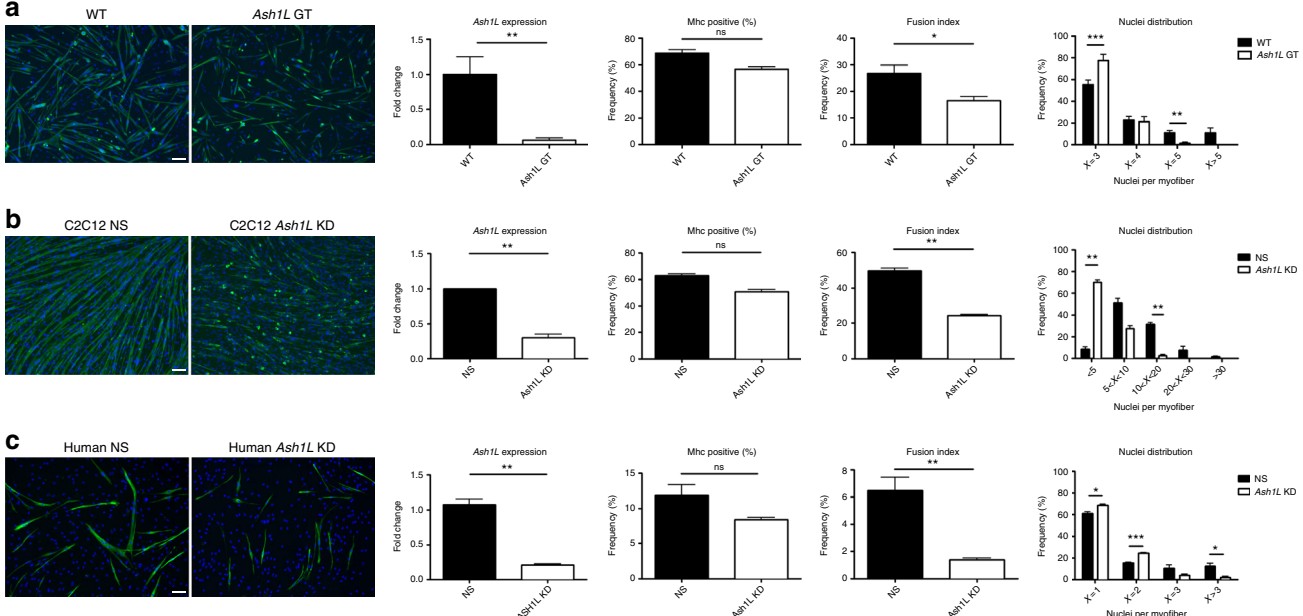

**Fig. 3** Effects of *Ash1L* ablation on myoblast fusion. **a** Primary myoblasts isolated from *Ash1L* GT mice display a fusion defect. Immunofluorescence for myosin heavy chain (in green) and nuclear staining (Hoechst) in primary cultures from wild-type (WT) and *Ash1L* GT mice at E18.5 stage, after inducing differentiation for 48 h. Scale bar, 100 μm. RT-qPCR analysis of *Ash1L* expression, validating *Ash1L* ablation in GT mice compared to wild-type mice. Percentage of Mhc-positive cells, calculated in comparison with the total number of nuclei. Fusion index, calculated as the number of nuclei present in myotubes (Mhc-positive and containing at least three nuclei) in comparison with the total number of nuclei. Nuclei distribution, calculated as the frequency of Mhc-positive cells containing the indicated number of nuclei. Unpaired two-tailed *t* test. Confidence intervals 95%. Results come from five biological replicates. **b** *Ash1L* knockdown leads to a muscle fusion defect. Immunofluorescence for myosin heavy chain and nuclear staining in C2C12 cells transfected with non-targeting (NS) and *Ash1L* siRNAs (*Ash1L* KD) and differentiated for 3 days. Scale bar, 100 μm. *Ash1L* expression, percentage of Mhc-positive cells, fusion index, and nuclei distribution evaluated as described above. Unpaired two-tailed *t* test. Confidence intervals 95%. Results come from three biological replicates. **c** Muscle fusion defect upon *Ash1L* knockdown is conserved in human. Immunofluorescence for myosin heavy chain and nuclear staining in human primary myoblast transfected with non-targeting (human NS) and *Ash1L* siRNAs (human *Ash1L* KD) and differentiated for 4 days. Scale bar, 100 μm. *Ash1L* expression, percentage of MHC-positive cells, fusion index, and nuclei distribution evaluated as described above. Unpaired two-tailed *t* test. Confidence intervals 95%. Results come from three healthy subjects. Source data are provided as a Source Data file. *$p \leq 0.05$, **$p \leq 0.01$, ***$p \leq 0.001$. NS not significant

formation in the absence of Ash1L. As shown in Supplementary Figure 3a, *Ash1L*-depleted cells did not present either proliferation or doubling time alterations compared to control cells. Time-lapse microscopy excluded also a significant migration impairment of *Ash1L*-KD myoblasts (Supplementary Figure 3b). Finally, the myogenic transcription factors MyoD and myogenin, and the muscle-specific marker desmin were expressed normally in *Ash1L*-KD cells (Supplementary Figure 3c), suggesting that Ash1L is not required to activate the muscle differentiation program. These findings imply that myoblasts depleted of Ash1L can proliferate, migrate, and activate muscle-specific gene expression, while are selectively defective in their ability to fuse.

**Identification of direct Ash1L targets in muscle cells**. In order to define the molecular mechanism through which Ash1L activates myoblast fusion, we performed RNA-sequencing comparing *Ash1L* KD to control C2C12 myoblasts. We identified 135 differentially expressed genes (false discovery rate (FDR) <0.05) (Supplementary Data 1). In line with the known involvement of Ash1L in transcriptional activation, most of the differentially expressed genes resulted downregulated (100/135, 74%), whereas only a few genes resulted upregulated (35/135, 26%). The downregulated genes returned several intriguing categories related to transcription activation, muscle differentiation, and cell adhesion (Fig. 4).

To determine which genes were directly impacted by Ash1L, we performed chromatin immunoprecipitation followed by

sequencing (ChIP-seq) to determine Ash1L genomic targets (Supplementary Data 2). Our analysis revealed extensive Ash1L occupancy at the gene bodies of its target genes with an enrichment near TSS (Fig. 5a, b), consistently with previous reports[2,34,60–62]. In the absence of chromatin state map (ChromHMM) data available for murine muscle cells, we compared our Ash1L peaks with chromatin state maps of available mouse tissues from ENCODE[63]. As expected, Ash1L resulted enriched at regions defined as active promoters or active enhancers (Fig. 5c). Intriguingly, we observed an enrichment of Ash1L also in regions characterized by the presence of the PcG-deposited histone mark H3K27me3 and defined as repressed, poised promoters, or poised enhancers (Fig. 5c).

Ash1L counteracts Polycomb activity through the deposition of H3K36me2[60,64–67]. As expected, by ChIP-seq we found that the levels of H3K36me2 were higher on expressed genes compared to not expressed genes as detected by RNA-sequencing (RNA-seq) in C2C12 (Mann–Whitney *p* value $< 2 \times 10^{-6}$; Fig. 5d). Intriguingly, among expressed genes the levels of H3K36me2 were significantly higher on the genes associated with Ash1L ChIP-seq peaks (Mann–Whitney *p* value $= 4.2 \times 10^{-11}$; Fig. 5d), supporting a role of Ash1L in their activation.

Next, we used gene set enrichment analysis (GSEA) to correlate the genes displaying Ash1L ChIP-seq peaks to the transcriptional downregulation upon *Ash1L* depletion. Leading-edge analysis allowed us to identify 45 direct Ash1L targets (normalized enrichment score (NES) = −2.35, FDR < 0.001) (Fig. 6a)

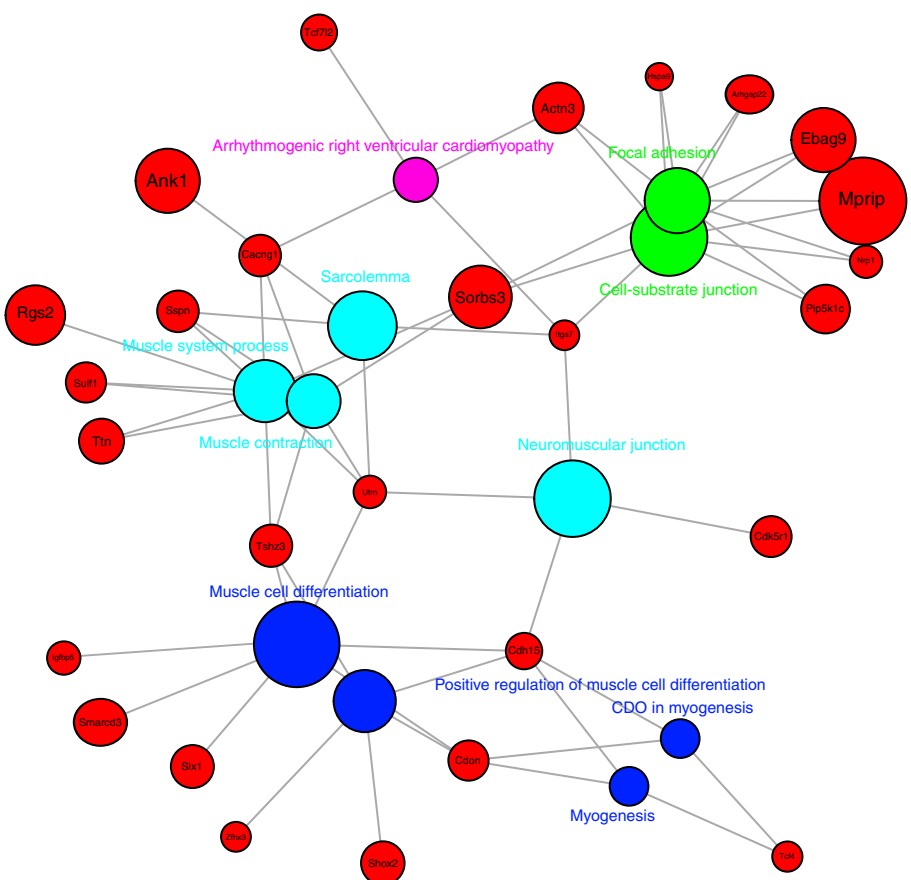

**Fig. 4** Pathways significantly downregulated by *Ash1L* ablation. Bipartite graph showing significantly enriched categories (KEGG, Biocarta, WikiPathways, Reactome, GO BP/MF/CC) (FDR <0.05, Z-score <−1.75, and n ≥ 3) and their associated genes within the 100 downregulated genes (FDR <0.1) in *Ash1L* KD myoblasts. The size of the gene nodes is proportional to the log 2 FC of downregulation, while the size of the category nodes is proportional to the significance of enrichment (−log 10 Adj. *p* value)

(Supplementary Data 3). Functional enrichment analysis returned an even stronger significance for categories involved with the positive regulation of myogenesis (Fig. 6b). Accordingly, Ash1L targets resulted also significantly enriched for genes activated during muscle differentiation ($p = 0.028$) (Supplementary Data 4). In line with the ability of Ash1L to positively regulate gene transcription by counteracting PcG-mediated silencing[52,60,64–66], ChIP enrichment analysis unveiled that the majority of Ash1L targets (39 out of 45) are also Polycomb targets (Supplementary Data 5).

In agreement with Ash1L-mediated activation, all the direct Ash1L targets tested were found significantly downregulated by *Ash1L* depletion in mouse and human muscle cells (Supplementary Figure 4a). Moreover, ChIP followed by real-time quantitative PCR (ChIP-qPCR) showed that *Ash1L* KD causes a specific reduction in Ash1L enrichment in the genomic regions of all these genes (Supplementary Figure 4b), strongly supporting the validity of our ChIP-seq results.

Collectively, our analyses promote Ash1L as a positive regulator of genes important for muscle formation.

**Ash1L promotes myoblast fusion through Cdon regulation.** To provide a possible molecular mechanism for the specific myoblast fusion defect observed upon *Ash1L* depletion, we inspected the list of direct Ash1L target genes. Among them, the only one with a known role in myoblast fusion was *Cdon*. The *Cdon* gene encodes for a cell-surface receptor mediating cell–cell interactions between muscle precursor cells and positively regulating myoblast

fusion[68]. Similar to mice lacking Ash1L, at E18.5 *Cdon*-knockout (KO) mice show smaller muscles compared to controls[68]. Moreover, like *Ash1L* GT myoblasts, *Cdon*-null primary myoblasts fail to fuse into myotubes[68]. We found that in vivo during muscle development or in response to muscle damage (Fig. 7a, b), and in vitro during muscle differentiation[69] (Fig. 7c), Cdon displays an expression pattern almost identical to that of Ash1L. Intriguingly, Cdon expression was significantly reduced in muscle tissue from *Ash1L* GT mice compared to wild-type mice, and in murine C2C12 or human primary muscle cells KD for *Ash1L* compared to controls (Fig. 7d, e). Our ChIP-seq results indicate a peak of Ash1L enrichment at the *Cdon* TSS (Fig. 7f). By ChIP-qPCR, we found a specific Ash1L enrichment near the *Cdon* TSS, which was significantly reduced compared to controls upon *Ash1L* KD (Fig. 7f). These results indicate that *Cdon* is a direct and specific Ash1L target.

To evaluate the biological relevance of *Cdon* activation by Ash1L, we expressed *Cdon* in *Ash1L*-KD muscle cells. After 3 days of differentiation, we performed immunofluorescence staining for Mhc and we evaluated both the fusion index and the number of nuclei per myofiber. While the fusiogenic capability of *Ash1L*-KD cells was significantly reduced compared to controls, cells depleted of *Ash1L* and concomitantly expressing *Cdon* were not significantly different from controls (Fig. 8). This evidence strongly supports that *Cdon* expression is sufficient to rescue the myoblast fusion defect caused by *Ash1L* ablation. Altogether, our findings indicate that *Cdon* is a key target gene activated by Ash1L to promote myoblast fusion.

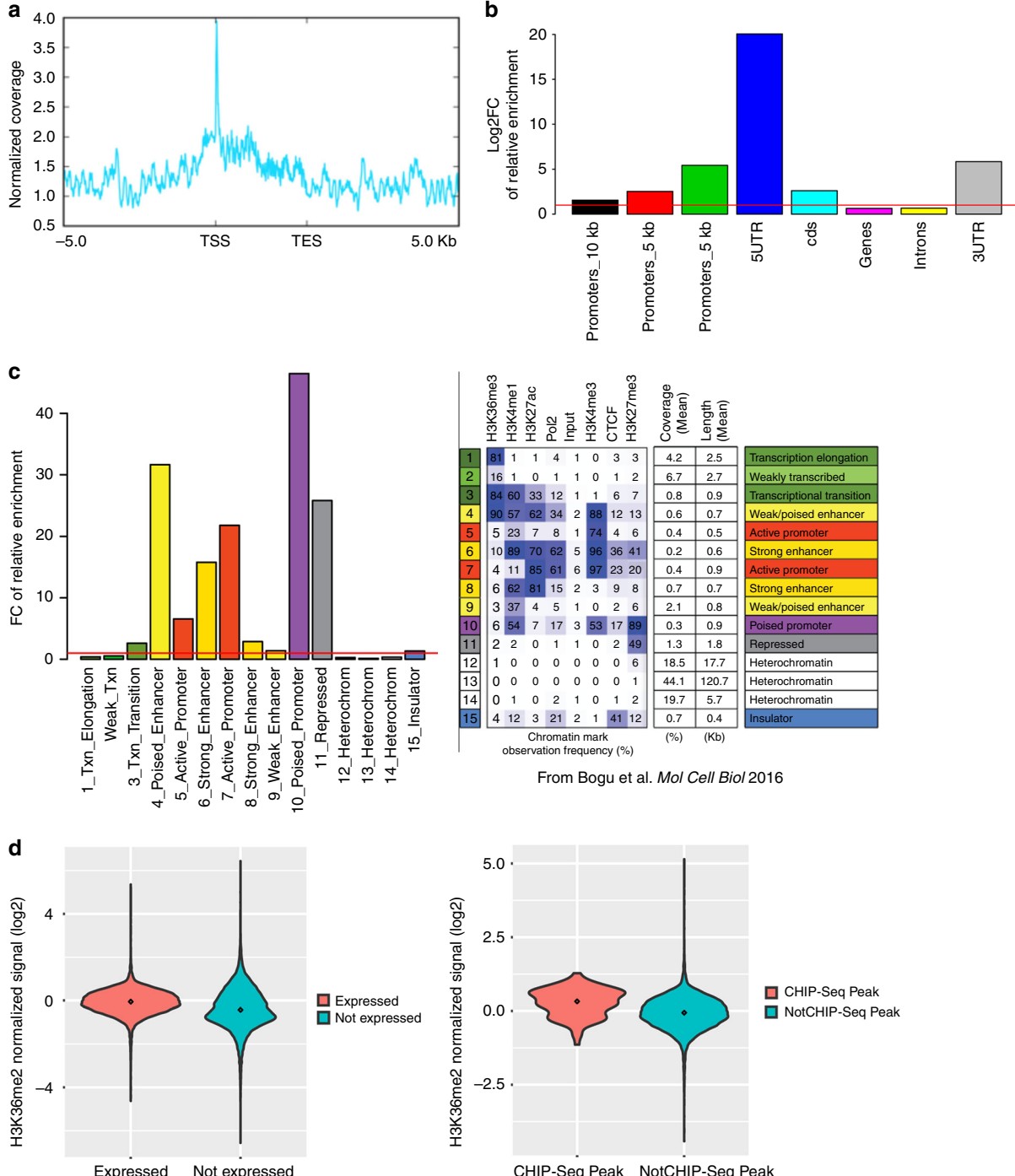

**Fig. 5** Distribution of Ash1L peaks and overlapping between Ash1L and Polycomb targets. **a** Ash1L ChIP-seq signal distribution over the scaled gene bodies (3 kb) of its targets showing enrichment around the TSS. One representative replicate of the three is shown. **b** Relative enrichment (log 2 FC) of Ash1l ChIP-seq peaks in classically defined genomic regions (promoters, 5'-UTR, CDS, genes, introns, and 3'-UTR). **c** Relative enrichment (FC) of Ash1L ChIP-seq peaks in genome-wide chromatin segments as found in Bogu et al.[63] (right panel reproduced with permission from American Society for Microbiology) in a union of mouse tissues using chromHMM tool. **d** Violin plots of normalized H3K36me2 signal over gene bodies of either expressed/unexpressed in myoblasts (left panel) or Ash1L peaks/not Ash1L peaks genes (right panel). Signal has been normalized against H3 ChIP control sample and three replicates averaged. Mann–Whitney $p$ value was used for comparison

## Discussion

Using a combination of cell biology, transcriptomics, genomics, and genetic analyses we show that Ash1L is evolutionarily required for myoblast fusion, while it is dispensable for myoblast proliferation, migration, and differentiation. In particular, our results suggest that muscle cells lacking Ash1L can differentiate and fuse to form the initial multinucleated myotube, but are defective in the further fusion of myoblasts with myotubes or myotubes with myotubes.

Myoblast fusion is a complex process required for muscle development, postnatal muscle growth, and adult muscle regeneration. Elegant studies in *Drosophila* and vertebrates have

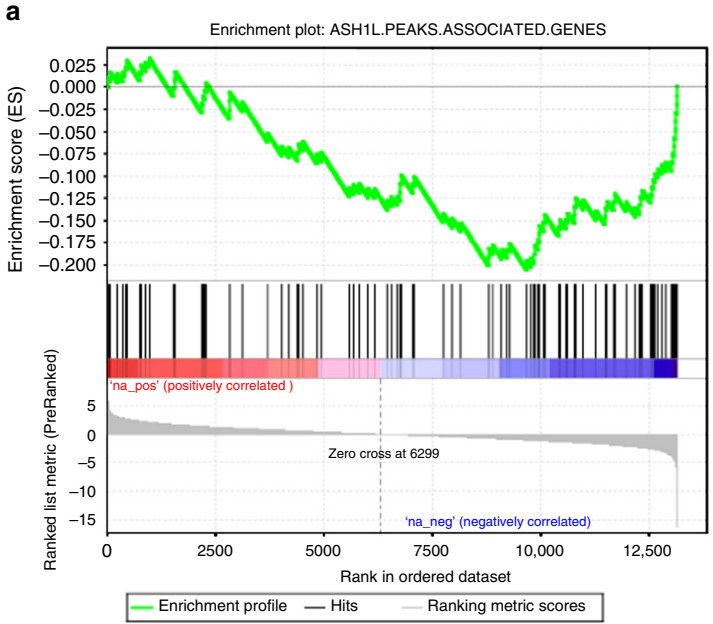

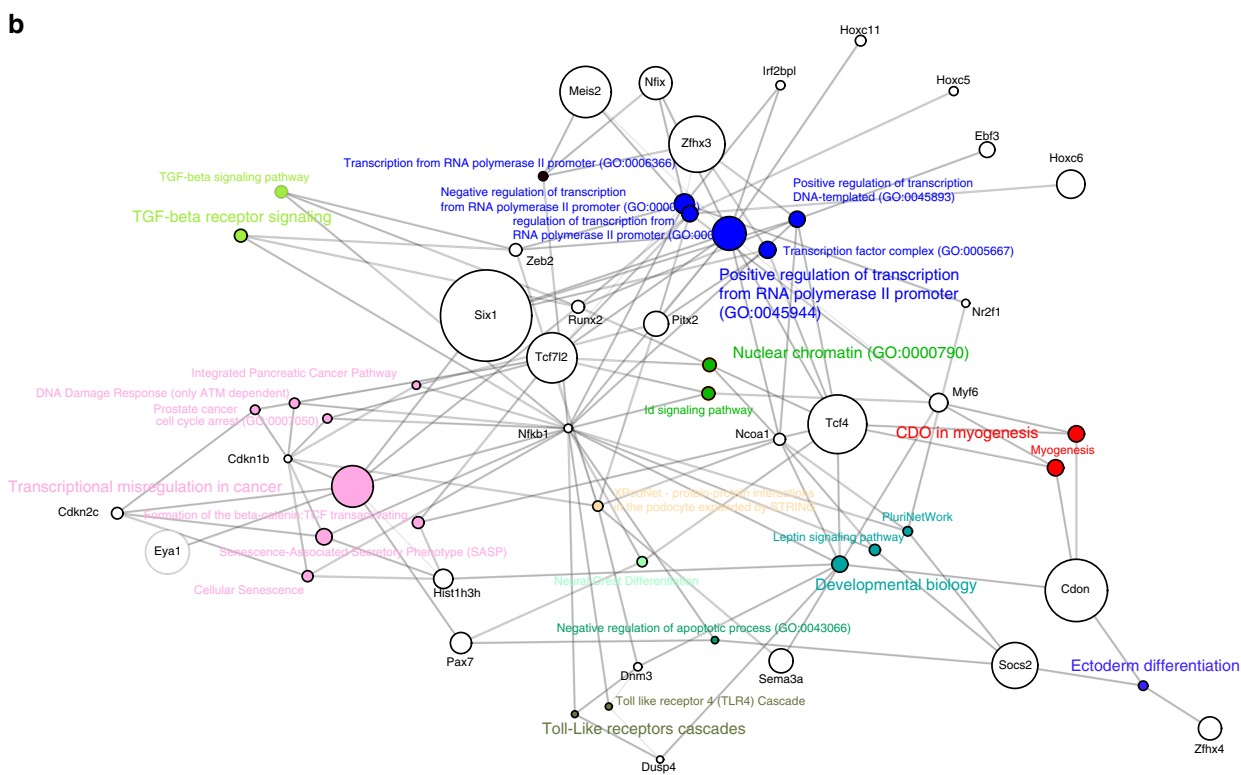

**Fig. 6** Identification of direct Ash1L targets. **a** Gene set enrichment analysis enrichment plot for Ash1L ChIP-seq peak-associated genes ($n = 146$, 96 expressed in myoblasts) obtained from the gene list pre-ranked according to the log 2 FC in myoblast KD vs. NS RNA-seq experiment. The leading edge analysis identified 45 genes as significantly downregulated and correlated to Ash1L binding. **b** Bipartite graph showing significantly enriched categories (KEGG, Biocarta, WikiPathways, Reactome, GO BP/MF/CC) (FDR <0.05, Z-score <−1.75, and $n \geq 3$) and their associated genes within the 45 direct targets identified with the GSEA analysis in **a**. The size of the gene nodes is proportional to the log 2 FC of downregulation, while the size of the category nodes is proportional to the significance of enrichment (−log 10 Adj. $p$ value)

identified many secreted, membrane, signaling, or cytoskeletal proteins belonging to the myoblast fusion machinery[8,10]. On the contrary, only few transcription factors involved in myoblast fusion are known[28–31]. Notably, no epigenetic regulator of myoblast fusion has ever been described.

Ash1L (Kmt2h) is a histone methyltransferase belonging to the Trithorax group. It positively regulates gene transcription by attaching two methyl groups to H3K36 and counteracting PcG-mediated gene silencing[32]. We found that the expression of *Ash1L* is positively correlated to myoblast fusion in vivo and in vitro. Intriguingly, *Ash1L* ablation causes a selective myoblast fusion defect in murine and human muscle cells. Accordingly, *Ash1L* GT mice display smaller muscles and fewer nuclei per myofibers likely due to a reduced myoblast fusion.

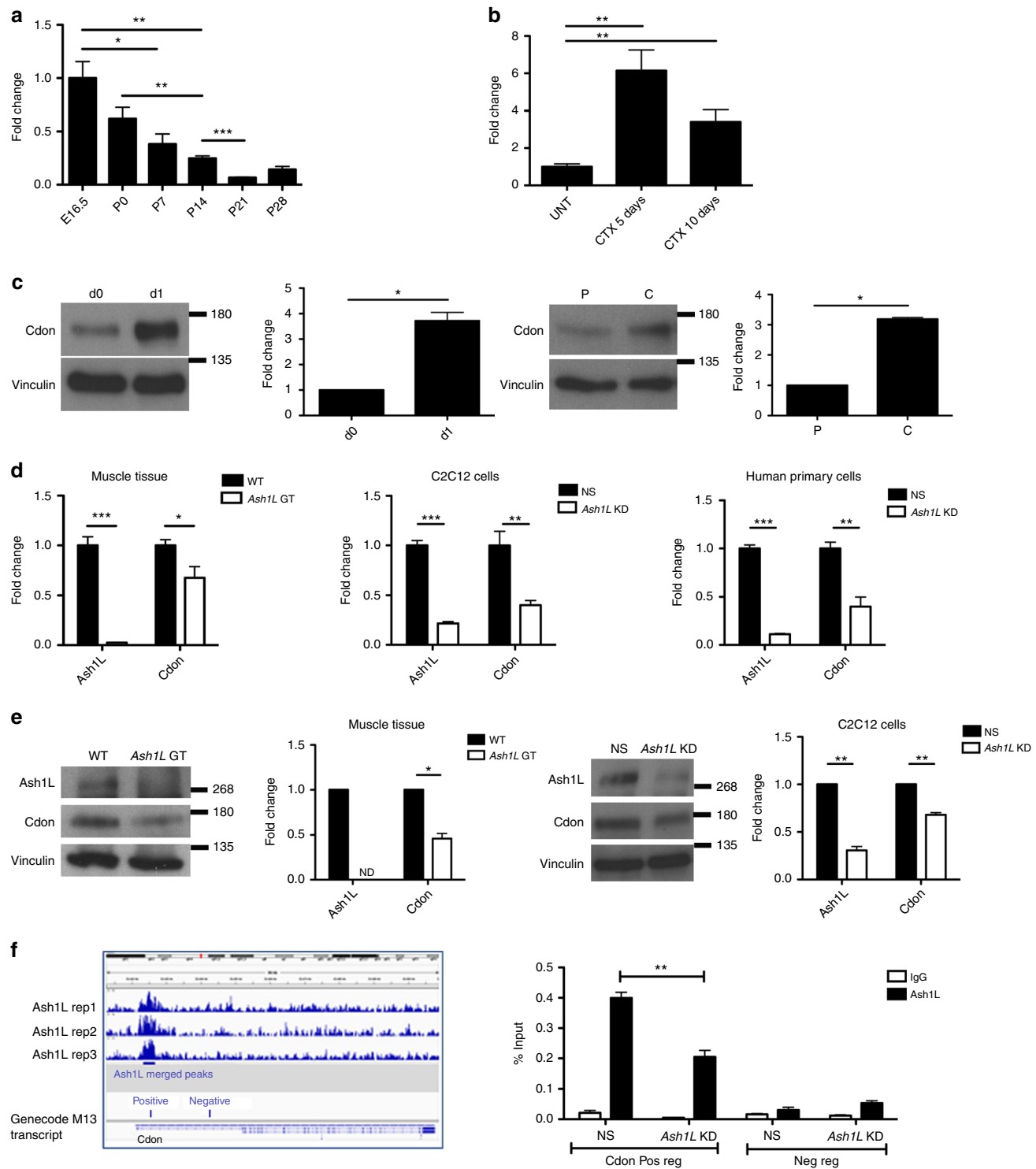

In agreement with its role in transcriptional activation, we found that Ash1L is enriched in active regions of the muscle cells genome, near the TSS of genes presenting also high levels of H3K36me2, and we observed mostly gene downregulation following Ash1L depletion. Genomic regions associated to Ash1L are also enriched for regulatory sequences that have been defined by the presence of the Polycomb-dependent histone mark H3K27me3[63]. Intriguingly, we found that the genes downregulated upon Ash1L-KD are enriched for known Polycomb targets. By combining RNA-seq and ChIP-seq, we identified 45 direct Ash1L targets. They are enriched for genes induced upon muscle differentiation and with an important role in myogenesis. Given the known role of Ash1L in counteracting Polycomb-repressive activity[32], it is tempting to speculate that the increase in Ash1L expression at the early phase of muscle differentiation could allow for the expression of specific Polycomb targets required for muscle differentiation.

Cell–cell contact is required for myoblast fusion and strongly stimulates myogenesis through the activation of promyogenic signal transduction pathways[21]. A key factor in cell contact-dependent signaling during myoblast fusion is Cdon, a multifunctional cell-surface protein serving as coreceptor for

**Fig. 7** The fusion gene *Cdon* is a direct target of Ash1L. *Cdon* expression positively correlates with *Ash1L* during muscle development (**a**) and regeneration (**b**). RT-qPCR analysis on muscle tissue from hindlimbs of mice from the embryonic stage E16.5 to adulthood and in tibialis anterior of wild-type adult mice, untreated (UNT), or 5 and 10 days after cardiotoxin (CTX) injection. Unpaired two-tailed *t* test. Confidence intervals 95%. $n = 6$ (**a**), $n = 4$ (**b**). **c** Cdon protein level during in vitro muscle differentiation and in proliferating myoblasts vs. confluent cells. Immunoblot of C2C12 cells at days 0 and 1, and densitometric analysis of Cdon signal relative to vinculin as housekeeping protein (left panel). Comparison between proliferating myoblasts (P) and confluent cells (C) (right panel). Paired two-tailed *t* test. Confidence intervals 95%. Data are the mean for three independent experiments. **d** RT-qPCR analysis comparing *Cdon* expression in the presence and absence of *Ash1L* in three different models: muscle tissue from hindlimbs of wild-type (WT) and *Ash1L* GT embryos at E18.5, C2C12 cells transfected with non-targeting (NS) or *Ash1L* siRNAs (*Ash1L* KD), and human primary cells transfected with non-targeting or *Ash1L* siRNAs, and collected at day 1 of differentiation. Unpaired two-tailed *t* test. Confidence intervals 95%. $n = 6$ (muscle tissue), four independent experiments (C2C12 cells), and three independent healthy subjects (human samples). **e** Western blotting analysis showing Cdon downregulation in muscle tissue from hindlimbs of *Ash1L* GT embryos at E18.5 compared to wild type (WT), and in C2C12 cells transfected with *Ash1L* siRNAs (*Ash1L* KD) compared to control (NS). Densitometric analysis of both Ash1L and Cdon signals relative to vinculin as housekeeping protein are reported on the right. Paired one-tailed *t* test. Confidence intervals 95%. $n = 3$ (muscle tissue), three independent experiments (C2C12 cells). **f** Ash1L specifically binds to *Cdon* genomic region. Genome browser representation of Ash1L ChIP-seq peaks (from three replicates), at *Cdon* genomic region. Relative positions of both positive and negative regions are shown at the bottom (left). Chromatin immunoprecipitation on C2C12 cells transfected with non-targeting (NS) or *Ash1L* siRNAs (*Ash1L* KD), and fixed at day 1 of differentiation, using Ash1L antibody and IgG as a negative control. qPCR analysis on positive and negative regions according to ChIP-sequencing results (right). Unpaired two-tailed *t* test. Confidence interval 95%. $n = 3$. Source data are provided as a Source Data file. $*p \leq 0.05$, $**p \leq 0.01$, $***p \leq 0.001$

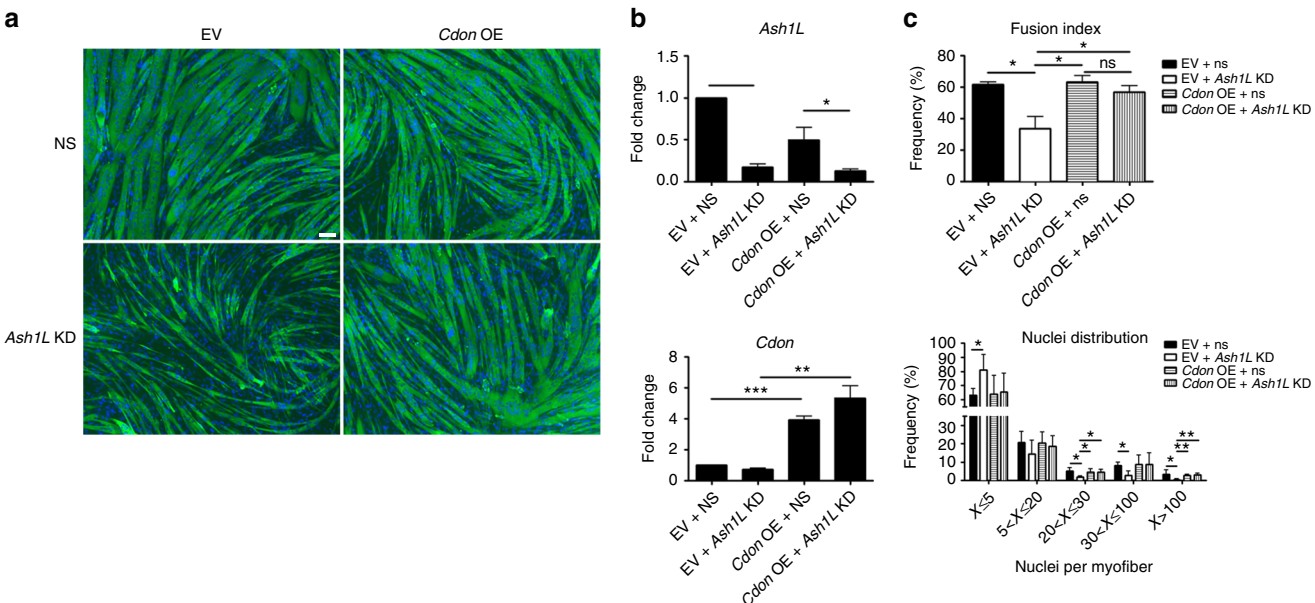

**Fig. 8** *Cdon* expression rescues fusion defects caused by *Ash1L* knockdown. **a** Immunofluorescence for myosin heavy chain and nuclear staining in C2C12 cells treated with two consecutive rounds of transient transfections: non-targeting (NS) or *Ash1L* siRNAs (*Ash1L* KD), and empty vector (EV) or *Cdon*-expressing vector (*Cdon* OE), before inducing differentiation for 3 days (left). Scale bar, 100 μm. **b** RT-qPCR analysis of *Ash1L* and *Cdon* expression in C2C12 cells transfected with: empty vector plus non-targeting siRNAs (EV + NS), empty vector plus *Ash1L* siRNAs (EV + *Ash1L* KD), *Cdon*-expressing vector plus non-targeting siRNAs (*Cdon* OE + NS), and *Cdon*-expressing vector +*Ash1L* siRNAs (*Cdon* OE + *Ash1L* KD). **c** Fusion index and nuclei distribution for each condition were calculated as described for Fig. 3. Unpaired two-tailed *t* test. Confidence intervals 95%. $n = 3$. Source data are provided as a Source Data file. $*p \leq 0.05$, $**p \leq 0.01$, $***p \leq 0.001$. NS not significant

N-cadherin, neogenin, Hedgehog, and Wnt pathways[21]. *Cdon* is a Polycomb target (Supplementary Data 5), is upregulated during muscle differentiation[70] and is the only direct Ash1L target gene with a known role in myoblast fusion[68]. We found that *Cdon* parallels *Ash1L* expression during muscle development and regeneration. Moreover, *Cdon* is significantly downregulated in muscles of *Ash1L* GT mice and upon *Ash1L* depletion in mouse and human muscle cells. *Cdon*-KO mice display smaller muscle fibers[68], as we observed in *Ash1L* GT mice. Furthermore, primary myoblasts isolated from both *Cdon*[68] and *Ash1L* GT mice display a myoblast fusion defect. Importantly, we found that *Cdon* expression is sufficient to rescue the myoblast fusion defect of *Ash1L*-KD cells indicating that (at least in vitro) *Cdon* is a key target through which Ash1L regulates myoblast fusion. Future

work is required to understand the relevance of the other direct Ash1L targets that we identified in muscle.

In vitro, myotube formation is generally obtained by growing myoblasts until near confluence in mitogen-rich medium, followed by a switch to mitogen-poor medium. Although the reduction of growth factors is an important differentiation signal, cell confluency itself is strongly promyogenic[21]. Intriguingly, we found that cell confluence is sufficient to strongly activate Ash1L expression even in the presence of a mitogen-rich medium. In the future, it would be interesting to determine which signaling pathway activated by high myoblast cell density is involved in Ash1L activation.

Ash1L is not a sequence-specific DNA-binding protein. Hence, the association of Ash1L to specific genomic regions is likely

mediated by gene-specific recruiters. We previously reported that the long non-coding RNA (lncRNA) *DBE-T* directly binds to Ash1L and recruits it to the FSHD muscular dystrophy locus for the activation of key disease genes[41]. By analogy, it would thus be interesting to find out whether additional lncRNAs are involved in Ash1L recruitment to its target genes identified in the present study.

The regulation of myoblast fusion is relevant for understanding the pathogenesis[15,17,53] and treatment of several muscle disorders. One of the muscular diseases in which myoblast fusion defects have been described is DMD[16,56,57]. Exon skipping with morpholino antisense oligonucleotides (PMO-AO) is a promising therapeutic strategy for DMD[71]. However, sporadic and variable rescue of dystrophin expression has been reported so far. It was recently shown that the fusion of PMO-loaded myoblasts to the repairing myofiber is a crucial aspect for PMO delivery into muscles of mdx mice[18]. Interestingly, we found a significant reduction of *Ash1L* expression in muscle tissue from both DMD patients and mdx mice. In line with a role of Ash1L in activating myoblast fusion and of its downregulation in DMD, it would be intriguing to test whether Ash1L co-delivery could increase the efficacy of PMO for the treatment of DMD.

In conclusion, we defined a key regulatory mechanism of myoblast fusion, a process required for muscle formation and repair. A better understanding of Ash1L-mediated regulation of myogenesis could contribute to clarify pathogenic mechanisms as well as to improve therapeutic approaches for muscle diseases.

## Methods

**Mouse handling**. All animal procedures were approved by the Institutional Animal Care and Use Committee of IRCCS San Raffaele Scientific Institute and were communicated to the Italian Ministry of Health and local authorities according to Italian law.

Wild-type C57BL/6J mice were purchased from Charles River (Calco, Italy) and sacrificed at E16.5, P7, P14, P21, and P28. Muscles from the entire hindlimb of these mice were dissected for RNA extraction and gene expression analysis. To induce muscle injury, 30 μl of CTX (0.1 mM) (Sigma) were injected into tibialis anterior muscles of 2-month-old wild-type males, using a 29 G syringe. Mice were sacrificed 5 or 10 days after injury.

*Ash1L* GeneTrap heterozygous mice were generated by using SIGTR ES cell line AL0395, carrying an insertion of pGT0lxr vector in *Ash1L* gene. We aggregated the ES cell line with 8-cell-stage embryos of the of C57BL/6J background, then the embryos were transplanted into oviducts of recipients of the ICR background to generate chimera mice. We performed more than 20 backcrosses in the C57BL/6J background. Homozygous mutant C57BL/6J embryos were generated by brother–sister heterozygous mating and analyzed at 18.5 days post coitum. *Ash1L+/+* from the same litter were used as control. Animals of both sexes were used for the study.

**Histology and immunofluorescence**. Hindlimbs of 18.5 embryos were dissected, frozen in isopentane, and cryosectioned at 8 μm thick.

Slices were fixed in 4% paraformaldehyde (PFA, Electron Microscopy Science) for 10 min, washed with phosphate-buffered saline (PBS), and stained with Harris hematoxylin for 5 min. After extensive washes with tap water, slices were stained with Gomori-trichrome solution (Cromotrope 2R 0.6%, Fast Green FCF 0.3%, phosphotungstic acid 0.6%, acetic acid (glacial) 1% (all from Sigma), pH 3.4) for 10 min, washed in tap water, and dehydrated with ethanol scale and xylene, and then mounted in xylene-based mounting medium.

Transverse sections were used to evaluate the CSA of myofibers. The analysis was performed using ImageJ software (http://imagej.net/Downloads)[72].

For myofibers nuclei counting, longitudinal sections were fixed in 4% PFA (Electron Microscopy Science), and then incubated with rabbit anti-laminin (Sigma, 1:300). Alexa Fluor 488 goat anti-rabbit was used as secondary antibody. Nuclei were stained by Hoechst 3342. Myonuclei in a single myofiber were counted and normalized to the length of the myofibers[72].

For Ash1L immunostaining on muscle sections, CTX-injured and control muscles of adult wild-type mice were cryosectioned at 8 μm thick. Transversal sections were fixed in 4% PFA (Electron Microscopy Science). Sections were permeabilized with Triton X-100 0.2% in PBS and antigen retrieval was performed using TE buffer, pH 9. Sections were then blocked and incubated with anti-Ash1L (Bethyl Laboratories; dilution 1:200). Alexa Fluor 488 goat anti-rabbit was used as secondary antibody. Nuclei were stained by Hoechst 3342.

For Ash1L immunostaining on C2C12, cells were incubated with CSK buffer (100 mM NaCl, 300 mM sucrose, 3 mM MgCl₂, 10 mM PIPES (pH 6.8)) 10 min on ice, washed with cold PBS, fixed in 4% PFA, and immunostained with Ash1L antibody (Bethyl Laboratories; dilution 1:200) followed by Alexa Fluor 488 goat anti-rabbit (Molecular Probes; dilution 1:500) and Hoechst (1 mg/ml; Sigma; dilution 1:2000).

Pictures were visualized and acquired using Imager.M2 or Observer.Z1 (Zeiss) microscopes. ImageJ software was used for the analysis. Adobe Photoshop C5 was used to compose the final pictures.

Antibodies are listed in Supplementary Table 1.

**Satellite cell-derived primary muscle cell cultures**. Cell preparations were collected from the entire hindlimb and forelimb of E18.5 embryos. The skin was removed, and the limbs were minced and digested at 37 °C with 1.15 mg/ml of collagenase type IV (Sigma) and 0.4 mg/ml dispase (Life Technologies) for 20 min. Tissue was filtered through 70 and 40 μm cell strainer and blood cells were lysated by ammonium chloride (Stem Cell Technologies). Cells were collected by centrifugation and resuspended in nutrient mixture F-10 Ham (Sigma) supplemented with 20% fetal bovine serum (FBS) (Hyclone), 1% penicillin–streptomycin, 1% glutamine (Gibco), 0.1% gentamicin (Sigma), and 5 ng/ml basic fibroblast growth factor (Preprotech). Then, cells were plated on collagen-coated dishes, after 30 min pre-plating in uncoated dishes (three incubations). Primary myoblasts were grown for 2 days and then differentiated in Dulbecco's modified Eagle's medium (DMEM; EuroClone) supplemented with 5% donor horse serum (EuroClone) for 48 h. Cells were collected and RNA extraction was performed using ReliaPrep RNA cell miniprep system (Promega).

**Mammalian cell culture**. All procedures involving human samples were approved by IRCCS San Raffaele Scientific Institute Ethical Committee. Human primary myoblasts from healthy donors were obtained from the University of Massachusetts Medical School Senator Paul D. Wellstone Muscular Dystrophy Cooperative Research Center for FSHD (http://www.umassmed.edu/wellstone/) in Worcester (MA, USA). Human primary myoblasts were grown in in 0.1% gelatin-coated dishes, in Ham's F-10 (Sigma-Aldrich) supplemented with 20% FBS, 0.5% chicken embryo extract, 1.2 mM CaCl₂ (Sigma-Aldrich), and 1% penicillin/streptomycin (100 U/ml final concentration; EuroClone). When 90% confluent, cells were switched to differentiation medium (4:1 DMEM:Medium 199 supplemented with 2% HS (donor horse serum; EuroClone), and 1% penicillin/streptomycin) for 4 days.

Murine C2C12 cells were cultured in DMEM supplemented with 10% FBS and 1% penicillin/streptomycin. For differentiation experiments, C2C12 cells were plated at 80% confluence in collagen-coated dishes and were differentiated for 3 days in DMEM containing 2% HS (EuroClone).

Human and murine muscle cells were routinely cultured in low oxygen (5%) conditions. For fusion index quantification, cells were fixed in 4% PFA and immunostained for Mhc with mouse MF20 antibody (Developmental Studies Hybridoma Bank; dilution 1:2) followed by Alexa Fluor 488 goat anti-mouse (Molecular Probes; dilution 1:500) and Hoechst (1 mg/ml; Sigma; dilution 1:2000). Samples were visualized using Observer.Z1 (N-Achroplan ×10/0.25 NA Ph 1) microscope (Zeiss). Fusion index analysis was performed with ImageJ by counting the number of nuclei belonging or not to myotubes (myosin-positive syncytia containing at least three nuclei). Nuclei number distribution was evaluated by counting the number of nuclei belonging Mhc-positive cells. At least three independent differentiation experiments were performed. For each experiment at least six fields were analyzed, counting at least 1000 nuclei for each cell type.

**Transient transfection**. For KD experiments, 90,000 cells were plated in each well of a 6-well plate. The day after, cells were transfected with small interfering RNAs (siRNAs) against *Ash1L* or non-targeting control siRNAs [SMARTpool: ON-TARGETplus ASH1L siRNA L-020460 (human), L-041891 (mouse) or Non-Targeting Pool D-001810; Dharmacon], using Lipofectamine 3000 reagent (Thermo Fisher Scientific) following the manufacturer's instructions. The day after, cells were placed in the differentiation medium. The day after, a second transfection with *Ash1L* or non-targeting control siRNAs was performed. Cells were fixed for immunofluorescence assay at day 3 of differentiation.

For rescue experiments, 50,000 cells were plated in each well of a 6-well plate. The day after, cells were transfected with *Ash1L* or non-targeting control siRNAs as above. The day after, proliferating cells were transfected with the pBABE-*Cdon* plasmid provided by Dr. Patrick Mehlen (University of Lyon, France) or pBABE-empty vector control using Lipofectamine LTX (Thermo Fisher Scientific). After overday incubation, the transfection medium was replaced with differentiation medium. Cells were fixed for immunofluorescence at day 3 of differentiation.

**Proliferation and migration assays**. For the proliferation assay, C2C12 cells were plated at the density of 2500 cells/cm². The day after, the transient transfection with siRNAs was performed as described above. Cells were counted every 9 or 15 h until 100% of confluence was reached. Doubling times were evaluated as follow: DT = (duration)Log(2))/((Log(final concentration) − Log(initial concentration)).

For the migration assay, C2C12 cells were transfected with siRNAs as described above. At the density of 60%, images were recorded every 5 min for 4 h, using Zeiss Axiovert S100 with Hamamatsu OrcaII-ER. Three fields with ×10 magnification

were analyzed for each condition, and 11 mononucleated cells were tracked for each field. The mean of the total distance covered by each cell of the field were evaluated and converted from pixel to μm. The results are expressed as mean of three independent experiments, for a total of 99 cells per condition.

**RNA extraction and RT-qPCR analysis**. Total RNA from muscle cells and tissues was extracted and treated with DNase 1, using RNA spin columns (PureLink RNA Mini Kit, Ambion) and Trizol reagent (Thermo Fisher Scientific), respectively. Complementary DNA (cDNA) was synthesized using Invitrogen's SuperScript III First-Strand Synthesis SuperMix. qPCRs were performed with SYBR GreenER qPCR SuperMix Universal (Invitrogen) using CFX96 Real-Time PCR Detection System (Bio-Rad). Relative quantification was calculated with CFX Manager Software V.1.6. Glyceraldehyde 3-phosphate dehydrogenase was used as house-keeping gene for sample normalization. Primers were designed using Primer3 tool (http://primer3.ut.eu) and are listed in Supplementary Table 1.

**Western blot analysis**. Protein extracts from C2C12 cells were obtained using Laemmli buffer 2×. Murine muscle tissue (hindlimbs from both wild-type and *Ash1L* GT embryos E18.5) was incubated in RIPA buffer (10 mM Tris-HCl, 1 mM EDTA, 0.5 mM EGTA, 1% Triton X-100, 0.1% sodium deoxicolate, 0.1% sodium dodecyl sulfate (SDS), 140 mM NaCl, 1 mM phenylmethanesulfonyl fluoride), then it was disrupted and homogenized using TissuLyser (Qiagen) for 3 min at 50 Hz, incubated 30 min on ice, and centrifuged at maximum speed 10 min at 4 °C. The supernatant was supplemented with Laemmli buffer.

Cell extracts were separated on 6 or 10% SDS-polyacrylamide gel electrophoresis gels, and then transferred to nitrocellulose membranes. Membranes were incubated with the antibodies listed in Supplementary Table 1. For detection, filters were incubated for 5 min with horseradish peroxidase chemiluminescent substrate (SuperSignal West Pico Chemiluminescent Substrate; Thermo Fisher Scientific). Antibody anti-vinculin was used as a loading control. Uncropped immunoblot images from representative experiments are shown in Supplementary Figure 5.

**Chromatin immunoprecipitation**. Chromatin from C2C12 cells (untreated or transiently transfected with siRNAs as previously described) was collected at day 1 of differentiation. Cross-linking was performed directly adding to the medium formaldehyde (Sigma-Aldrich) to the final concentration 1% and incubating cells with gentle swirl 10 min at room temperature (RT). After formaldehyde quenching with 125 mM glycine (Sigma-Aldrich) for 5 min, cells were washed with PBS and harvested by scraping and pelleted. Each cell pellet derived from one 15 cm dishes was lysed in 1 ml of LB1 solution (50 mM HEPES-KOH, pH 7.5, 140 mM NaCl, 1 mM EDTA, 10% glycerol, 0.5% NP-40, 0.25% Triton X-100; all from Sigma-Aldrich) for 10 min on ice. The samples were centrifuged at $1350 \times g$ for 5 min at 4 °C. The resulting pellet was washed in 1 ml of LB2 solution (10 mM Tris-HCl, pH 8; 200 mM NaCl, 1 mM EDTA, 0.5 mM EGTA; all from Sigma-Aldrich) with gentle swirl 10 min at RT. Next, samples were centrifuged at $1350 \times g$ for 5 min at 4 °C and the resulting pellet was lysed in 1 ml of LB3 solution (10 mM Tris-HCl, pH 8; 100 mM NaCl, 1 mM EDTA, 0.5 mM EGTA, 0.1% Na-deoxycholate, 0.5% N-laurylsarcosine; all from Sigma-Aldrich). LB1, LB2, and LB3 solutions were supplemented with protease inhibitor (Complete EDTA-free Protease Inhibitor Cocktail Tablets; Roche). Lysates were sonicated with Bioruptor (Diagenode). Briefly, 1 ml aliquots of LB3 lysates in 15 ml polystyrene tubes were sonicated for two rounds of 10 min each (high intensity, 30 s on–30 s off). Before the precipitation step, Triton X-100 (Sigma-Aldrich) was added to chromatin samples at a final concentration of 1%. For Ash1L and its control, 50 μg of chromatin were diluted in a final volume of 800 μl in LB3 buffer and incubated with 10 μg of antibody overnight at 4 °C in rotation. For both histones and histone modifications, 25 μg of chromatin plus 5 μg of antibody were used. The equivalent of 10% of single IP was collected as input fraction.

The day after, Dynabeads protein G (Thermo Fisher Scientific) were washed three times with 0.5% bovine serum albumin (BSA) (Jackson Immunoresearch) in PBS and 50 μl of washed beads were added to each sample and let for 3 h at 4 °C in rotation. Then, five washes with buffer 1 (2 mM EDTA, 20 mM Tris-HCl pH 8, 1% Triton X-100, 150 mM NaCl, 0,1% SDS; all from Sigma-Aldrich) and four washes with buffer 2 (2 mM EDTA, 20 mM Tris-HCl, pH 8, 1% Triton X-100, 500 mM NaCl, 0.1% SDS; all from Sigma-Aldrich) were performed. Before the elution step, an additional wash with TE buffer (10 mM Tris-HCl, pH 8, 1 mM EDTA; all from Sigma-Aldrich) was performed. Next, samples were centrifuged for 2 min at $1000 \times g$ at 4 °C. The supernatant was discarded and 250 μl of elution buffer (TE buffer with 2% SDS) were added to the beads–antibody–chromatin complex. Samples were incubated in a thermo mixer for 15 min at 65 °C while shaking, and then centrifuged for 1 min at RT at $16,360 \times g$. The supernatant was transferred to a new tube and samples, together with the input fractions to which 170 μl of elution buffer were added (final volume 250 μl), were cross-link reverted overnight at 65 °C. The day after, samples were incubated 1 h at 55 °C with 5 μl of Proteinase K (stock 20 mg/ml; Promega). For DNA purification, the QIAquick PCR Purification Kit was used (Qiagen), following the manufacturer's recommendations. DNA was eluted in 50 μl of TE buffer and 1 μl was analyzed in quantitative real-time PCR with the Sybr GreenER qPCR Kit (Thermo Fisher Scientific). Primers and antibodies used for ChIP-qPCR analysis are listed in Supplementary Table 1.

**RNA-sequencing**. Total RNA was extracted using RNA spin columns (PureLink RNA Mini Kit, Ambion). RNA quantification was carried out using Qubit Fluorimetric Quantification (Life Technologies) and 500 ng of RNA were used to prepare the library. Library generation was performed using QuantSeq 3′ mRNA-Seq Library Prep Kit. The procedure was initiated by oligo-dT priming, then the first-strand synthesis, and RNA removal was followed by random-primed synthesis of the complementary strand. The resulting double-stranded cDNA was purified with magnetic beads, in order to remove all reaction components. The adapter sequences were added by library PCR amplification, the proper cycle number was previously determined by qPCR. The library was purified from PCR components using purification beads. Finally, the quality of the samples was determined using Bioanalyzer (Agilent Technologies).

Libraries were finally sequenced using an Illumina HiSeq 2500 system based on standard protocols (100 bp single end).

Raw sequencing reads (FASTQ) were processed individually and mapped to the mouse genome reference version GRCm38 (mm10) from Gencode (M13). The mapping was performed using STAR (v2.5.3a) (https://github.com/alexdobin/STAR/) using soft clipping and all other parameters set to default values according to recommended data analysis workflow by Lexogen. Gene abundance was determined using featureCounts (http://bioconductor.org/packages/release/bioc/html/Rsubread.html) and normalized using calcNormFactors function from edgeR R/Bioconductor package (http://bioconductor.org/packages/release/bioc/html/edgeR.html) followed by differential gene expression analysis using LIMMA (http://bioconductor.org/packages/release/bioc/html/limma.html) on a design including myoblasts and myotubes for both NS and *Ash1L* KD ($n = 3$ each group). Genes were considered as expressed when showing more than 1 cpm (count per million) and differentially expressed when showing an FDR < 0.1.

**ChIP-sequencing**. ChIP samples from C2C12 cells at day 1 of differentiation were obtained under conditions described above. DNA was further sonicated after elution, using the Ultra-Sonicator E220 (Covaris) at 10 V for 90 s, and the fragments sizes were evaluated by the Bioanalyzer (Agilent Technologies). ChIP-seq libraries were prepared using the Accel-NGS 2S Plus DNA Library Kit (Swift Biosciences), according to the protocol for DNA <10 ng. During Ligation I step, a unique indexed adapter (SI-ILM2S) was added to label each library. After the post-ligation II SPRI phase, a qPCR amplification was performed using KAPA Library Quantification Kit (Illumina Platforms), to determine the average size-adjusted concentration of the samples. An extra step of amplification was performed at 12 cycles for all the samples with the exception of input and total H3, to reach a proper concentration for sequencing.

Libraries were sequenced by Macrogen biotechnology company (www.macrogen.com/eng/) using Illumina HiSeq and analyzed with standard pipelines. Briefly, reads were checked for quality parameters with FastQC (ref) and trimmed for adapter content and base quality (≥30). They were aligned to the mouse genome (mm10) with BWA-MEM (https://arxiv.org/abs/1303.3997) and PCR duplicates were removed with Picard tools (https://broadinstitute.github.io/picard/). Properly paired and uniquely mapped reads were additionally cleaned by the removal of blacklisted regions (https://www.encodeproject.org/) and finally Ash1L binding narrowPeaks and broadPeaks were called for the three triplicate samples with MACS2 callpeak software (https://github.com/taoliu/MACS) ($q$ value < 0.05, slocal 2000) using the Input sample as control. Broad peaks called ($q$ < 0.05) in either of the three replicates were merged using bedTools (https://bedtools.readthedocs.io/en/latest/) and annotated to genes by their distance to TSS using the ensemble annotation (version_oct2016) and R/Bioconductor package (ChIPpeakAnno) (http://bioconductor.org/packages/release/bioc/html/ChIPpeakAnno.html).

**Enrichment of Ash1L peaks**. Enrichment of Ash1L peaks detected in the ChIP experiment was made firstly against classically defined genomic regions (promoters, 5′-untranslated region (UTR), 3′-UTR, coding sequence, introns) (Fig. 5b). The different regions were downloaded from the UCSC Genome Table Browser (mm10) and the intersection was calculated using bedtools intersect (at least a 10% of intersection between the regions and the peaks). The relative enrichment was calculated by dividing the Ash1L peaks overlap relative fractions of the different regions by their equivalent genome-wide relative fractions.

For the chromHMM enrichment in Fig. 5c, a union of all chromatin segments for the available mouse tissues from ENCODE[63] was used to find the overlap with Ash1L peaks because of the absence of mouse muscle tissue chromatin segmentation data. The overlap and relative enrichment were then calculated in the same way as for the classically defined regions in Fig. 5c.

The profile of Ash1L ChIP-seq coverage around the TSS was calculated by using scaled regions (to 3 kb) of the 45 direct targets of Ash1L as determined by ChIP-seq and RNA-seq using deeptools2 plotProfile tool (https://deeptools.readthedocs.io/en/develop/).

**H3K36me2 ChIP-seq analysis**. Reads coming from H3K36me2 ChIP-seq were processed similarly to Ash1L ChIP-seq reads. The levels of H3K36me2 methylation were extracted from the alignments using BedTools (https://bedtools.readthedocs.io/en/latest/)]. They were then normalized to library size, genome size, and to the anti-H3 IP control sample levels and averaged between three replicates. The levels

shown in Fig. 5d were calculated using the mouse annotation file from gencode (all genes) or after filtering it for genes expressed in myoblasts as determined by the RNA-seq experiment (expressed genes) or genes annotated with an Ash1L peak as detected by the Ash1L ChIP-seq experiment (ChIP-seq peaks). The gene body was considered from the 5′ end of the most upstream transcript to the 3′ end of the most downstream transcript of the gene as present in the current annotation. Violin plots were used using ggplots2 R package.

**Gene set enrichment analysis.** GSEA (http://software.broadinstitute.org/gsea/index.jsp) used GSEAPreranked with default parameters using the log 2 fold changes calculated by LIMMA (http://bioconductor.org/packages/release/bioc/html/limma.html) on the Ash1L-KD effect (KD vs. NS) in myoblasts and the set of 146 genes identified as potential Ash1L targets by ChIP-seq.

**Functional enrichment.** Enrichr web tool (http://amp.pharm.mssm.edu/Enrichr/) was used to test for functional enrichment with special focus on KEGG, Biocarta, Reactome, and Wikipathways databases (2016) for pathway enrichment and Gene Ontology Biological Processes, Molecular Function, and Cellular Components (2015) for ontologies. ChEA (ChIP-Seq Enrichment Experiment Database, 2016) was used to explore enrichment in DNA-binding factors. Categories were considered enriched when shown an FDR <0.05, a $Z$-score <−1.75 and 3 or more genes in overlap.

LIMMA analysis (http://bioconductor.org/packages/release/bioc/html/limma.html) through GEO2R was used to test for activation of Ash1L targets during muscle differentiation of C2C12 cells. The following conditions were compared: 90% confluency vs. 50% confluency and day 1 vs. 50% confluency in GSE11415; 24 vs. −24 h and 0 vs. −24 h in GSE17039; day −1 vs. day −2 and day 2 vs. day −2 in GSE989. All comparisons were done in triplicate. In each comparison the probe sets that were upregulated and downregulated with FDR <0.05 were identified. A gene was defined as upregulated if at least one of its probe sets was upregulated in one comparison and if the number of its probe sets upregulated across all conditions was greater than the number of downregulated ones. Then, a Fisher exact test was used to evaluate the enrichment of upregulated genes in Ash1L targets, with respect to genes that have an Ash1L ChIP peak, but do not show differential expression in our experiments.

**Statistical analysis.** Unless stated otherwise, statistical comparisons were two-tailed tests and performed using GraphPad Prism5.0a (GraphPad Software). The type of statistical test and the number of independent experiments is provided for each dataset in the corresponding figure legend. The differences were considered statistically significant when $p \le 0.05$ and was reported as follows: ****$p \le$ 0.0001, ***$p \le 0.001$, **$p \le 0.01$, *$p \le 0.05$.

## Data availability
The source data underlying Figs. 1a, 1b, 1c, 1d, 2a, 2b, 3a, 3b, 3c, 7a, 7b, 7c, 7d, 7e, 7f, 8b, 8c and Supplementary Figures 1, 2, 3a, 3b, 3c, 4a and 4b are provided as a Source Data file. RNA-seq of myoblasts and myotubes in control and Ash1L-knockdown muscle cells and ChIP-seq for Ash1L and H3K36me2 and controls have been deposited in the GEO database under accession code GSE110957. The muscle tissue differentiation datasets GSE11415, GSE17039, GSE989 were re-analyzed after downloading from GEO database. Other data and materials are available from the corresponding author upon reasonable request.

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

## Acknowledgements

Work in the Gabellini lab was supported by the Italian Epigenomics Flagship Project, the FSHD Global Research Foundation, the ERA-Net for Research on Rare Diseases, the Italian Ministry for Health, the FSH Society, and the Associazione Italiana per la Ricerca sul Cancro (AIRC IG 19919). We thank Giovanni Tonon and Gabellini lab members for helpful discussion. We thank Drs. Susumu Hirose (National Institute of Genetics, Japan) and Haruhiko Koseki (RIKEN IMS, Japan) for support in the generation of *Ash1L* GeneTrap mice. We thank the University of Massachusetts Medical School Senator Paul D. Wellstone Muscular Dystrophy Cooperative Research Center for FSHD for the primary human myoblasts. We also thank Dr. Patrick Mehlen (University of Lyon, France) for the pBABE-*Cdon* plasmid.

## Author contributions

I.C., R.C., and G.F. performed experiments and analyzed data. G.C. and K.N. provided mice and discussed the results. J.M.G.-M. and I.M. performed bioinformatics and discussed the results. D.G. analyzed data and supervised the work. I.C., J.M.G.-M., I.M., and D.G. wrote the manuscript.

## Additional information

**Competing interests:** The authors declare no competing interests.

