## [Peer Review File · Nature Communications]

Reviewers' Comments:

Reviewer #1:

Remarks to the Author:

In this manuscript, Castiglioni et al. describe and explore a new function for the Trithorax group gene Ash1l in myoblast fusion and muscle differentiation. Their findings are based on observations in C2C12 cells (in which most mechanistic investigations were performed), but also concordant findings in genetically engineered mice with an Ash1l "gene trap" loss-of-function allele, some data using primary human myoblasts and gene expression analysis in mice and patients with muscular dystrophy. The authors report high Ash1l expression in muscle at stages in which myoblast fusion is known to happen in mice (i.e. perinatally), as well as increased expression in muscle after chemical injury and in C2C12 cells upon induced differentiation. They then provide evidence for decreased myoblast fusion in the absence of Ash1l *in vivo* and *in vitro*. They go on to explore genomic targets of Ash1l by ChIPSeq and RNASeq, eventually focusing on Cdon, as candidate downstream target with features of a direct target and known functions in myoblast fusion. Using transfection in C2C12 cells, the authors provide data suggesting that Cdon expression can rescue loss of Ash1l in C2C12 cell differentiation.

Altogether, this is an interesting paper providing a new observation with potential significance for our understanding of muscle development, regeneration and disease. The manuscript provides for the first time a characterization of the putative molecular targets of Ash1l in muscle, and an attempt at dissecting key downstream mechanisms. Experiments seem to be generally well performed. Some concerns remain, however, and the paper could be improved by addressing them:

1) The ChIP-Seq data could be improved further by providing parallel data in C2C12 cells depleted of Ash1l, as provided for ChIP-PCR in fig. 5F. Such data would enhance the confidence with which positive Ash1l binding peaks can be called. Do the authors have access to such data?

2) Some aspects of the rescue experiment presented in Fig. 6 are problematic. Indeed, the percentage of myotubes with $n < 5$ nuclei in this experiment appears to be around 60%, as compared to only about 10% in a similar experiment presented in Fig. 3B. Ash1l knockdown increases the percentage of myotubes with $n < 5$ nuclei in both experiments, but the dynamic range is much smaller in Fig. 6 than in Fig. 3B, and thus conclusions appear to be based on much smaller absolute differences in the latter experiment. This does call into question the conclusion that Cdon overexpression can indeed fully rescue the consequences of Ash1l loss. There could be technical reasons for the findings, and may be the double transfection is an issue, but it would be great if the authors could address this issue. In addition, Cdon overexpression appears to decrease Ash1l expression – is this a consistent finding?

3) The authors refer to Ash1l KO mice throughout the paper when in fact they appear to be working with a "gene trap" allele and not a complete knockout. The nomenclature should be clarified.

4) The authors comment on perinatal lethality in Ash1l-deficient mice. They appear to use the same gene trap allele from the Sanger Institute that has been described before by others (see for example Brinkmeier et al., Biol Reprod 2015). In this other work, there were a significant fraction of homozygous gene trap mice that survived into adulthood, although not at a mendelian ratio. Do the authors have more information about rates of perinatal lethality in their hands? Could this be related to the degree of backcrossing of the different strains? Are there other possible explanations?

Minor comments:

- In Fig. 2, primary data from longitudinal sections would be very useful.
- Line 150: please clarify in the text in which setting RNA-Seq was performed, rather than referring simply to "muscle cells".
- Fig. 1: panels D and E are mislabeled in the legend.
- For clarity, Fig. 6 would benefit from the creation of subpanels.
- Lines 138-143: panels of fig. S1 are incorrectly referred to.

Reviewer #2:

Remarks to the Author:

In this manuscript, the authors identified up-regulation of Ash1L during myoblast fusion and regenerating skeletal muscle. Ash1L is a histone methyltransferase responsible for H3K36me2 and a member of the Trithorax proteins counteract with Polycomb proteins which are a transcriptional repressor. Ash1L is known to be involved in FSHD, autism and cancer. Gene knockout and gene knockdown experiments showed that Ash1L is required for proper myoblast fusion in mouse primary myoblast, C2C12 cells and human primary myoblast cultures. Ash1L KO fetal muscle showed the reduced fiber diameters. RNA-seq and ChIP-seq analyses demonstrated the 34 direct target genes for Ash1L. Among them, the authors also found Cdon which has been reported to be the essential role in myoblast fusion. Indeed, Ash1L can activate Cdon gene expression and overexpression of Cdon can rescue the myoblast fusion defects seen in Ash1L gene knockdown in C2C12 cells. Therefore, the authors concluded that Ash1L is the first epigenetic factor for myoblast fusion.

The data presented in this manuscript is very clear and interesting. However, this reviewer has some concerns with the expression of Ash1L and Ash1L KO mouse phenotypes since Ash1L KO mouse has been reported previously.

Major issues:

1. The author mentioned that Ash1L gene KO mice generated by SIGTR ES cell line AL0395 are perinatal lethal. However, the published data utilizing the same strain of mice (Brinkmiere et al, Bio Reprod, 93(5), 121, 2015) showed normal mendelian distribution until postnatal day 7 (p7) and the number of homozygous mice declines by p21. However, around half of the homozygous mice are still alive until adult. The authors did not cite this paper. Did you see any survived Ash1L mice postnatally? If not, what is the difference between the published data and this manuscript?
2. It is important to show immunostaining for Ash1L during myoblast culture and muscle regeneration after CTX injection to see whether Ash1L protein is upregulated during myoblast fusion in the nucleus.
3. Previous work (Cole et al., Dev Cell, 7(6), 843, 2004) Cdon enhances MyoD activity through promotion of MyoD-E2A heterodimerization. Please examine whether Ash1L also promotes MyoD activity and loss of function of Ash1L reduces this activity.
4. Cdon was downregulated by loss of Ash1L in both primary myoblasts and C2C12 cells. Please show Cdon expression by immunostaining. In addition, please examine whether Cdon expression in skeletal muscle in both wild-type and Ash1L KO mice.

Minor issues:

1. The authors described in the Methods section that Ash1L GeneTrap heterozygous mice, were generated by using SIGTR ES cell line AL0395, carrying an insertion of pGT01xr vector in Ash1L gene. Were these mice generated by authors using SIGTR ES cell line AL0395? Or did authors obtain mice from an organization such as Jackson Laboratories?

REVIEWER#1

The Reviewer found our study “*an interesting paper providing a new observation with potential significance for our understanding of muscle development, regeneration and disease. The manuscript provides for the first time a characterization of the putative molecular targets of Ash1l in muscle, and an attempt at dissecting key downstream mechanisms. Experiments seem to be generally well performed.*”

We are grateful to the Reviewer for her/his positive comments about the novelty and relevance of our work.

“*Some concerns remain, however, and the paper could be improved by addressing them:*

- 1) *The ChIP-Seq data could be improved further by providing parallel data in C2C12 cells depleted of Ash1l, as provided for ChIP-PCR in fig. 5F. Such data would enhance the confidence with which positive Ash1l binding peaks can be called. Do the authors have access to such data?”*

Unfortunately, we don't have access to ChIP-seq data of C2C12 cells depleted of Ash1L. To generate these data, we would have to perform ChIP-seq for Ash1L and control IgG in Ash1L knockdown and control knockdown C2C12 cells, all in biological triplicate. This will entails 12 new ChIP-seq experiments in addition to the 6 ChIP-seq experiments that we have already reported in the manuscript. This will be a very expensive experiment, which will require several months to be completed and analyzed. Thus, to enhance the confidence with which positive Ash1L binding peaks have been called, we decided to perform ChIP-qPCR for additional Ash1L direct targets similarly to what already shown in Figure 5F for Cdon. Remarkably, for all genes tested we obtained results in line to what shown in Figure 5F strongly supporting the validity of our ChIP-seq results.

2) *Some aspects of the rescue experiment presented in Fig. 6 are problematic. Indeed, the percentage of myotubes with $n < 5$ nuclei in this experiment appears to be around 60%, as compared to only about 10% in a similar experiment presented in Fig. 3B. Ash1l knockdown increases the percentage of myotubes with $n < 5$ nuclei in both experiments, but the dynamic range is much smaller in Fig. 6 than in Fig. 3B, and thus conclusions appear to be based on much smaller absolute differences in the latter experiment. This does call into question the conclusion that Cdon overexpression can indeed fully rescue the consequences of Ash1l loss. There could be technical reasons for the findings, and may be the double transfection is an issue, but it would be great if the authors could address this issue. In addition, Cdon overexpression appears to decrease Ash1l expression – is this a consistent finding?*

As the Reviewer suggests, it is difficult to compare the absolute numbers of Fig. 3B and Fig. 6. Indeed, there are several technical reasons accounting for the non-identical results. In Fig. 3B, we started with cells plated at relatively high density and performed two consecutive transfections with Ash1L or control siRNAs using Lipofectamine 3000 reagent: the first while the cells were in growth medium and the second two days after when the cells were already in differentiation medium. In Fig. 6, we plated the cells

relatively sparse and we performed a single transfection with Ash1L or control siRNAs using Lipofectamine 3000, and the following day we performed the transfection with the Cdon or control plasmids using Lipofectamine LTX reagent. Both transfections were performed in growth medium. Importantly, despite some difference in nuclei distribution due to the above described technical issues, the results of Fig. 6 strongly indicate that activation of Cdon expression is a key mechanism through which Ash1L stimulates myoblast fusion. We modified the Material and Methods section to better explain the technical differences between the experiments of Fig.3B and Fig. 6. On a separate note, by re-evaluating all our experiments, we found that expression of Ash1L is variably but non-significantly affected by Cdon overexpression.

- 3) *The authors refer to Ash1l KO mice throughout the paper when in fact they appear to be working with a “gene trap” allele and not a complete knockout. The nomenclature should be clarified.*

The Reviewer is correct. We apologize for overlooking this issue and corrected the nomenclature throughout the text and Figures.

- 4) *The authors comment on perinatal lethality in Ash1l-deficient mice. They appear to use the same gene trap allele from the Sanger Institute that has been described before by others (see for example Brinkmeier et al., Biol Reprod 2015). In this other work, there were a significant fraction of homozygous gene trap mice that survived into adulthood, although not at a mendelian ratio. Do the authors have more information about rates of perinatal lethality in their hands? Could this be related to the degree of backcrossing of the different strains? Are there other possible explanations?*

Brinkmeier et al. reported the expected Mendelian distribution of genotypes (1:2:1) through Postnatal Day 7 (P7). Ash1L GT/GT mutants were significantly underrepresented after P7 and older. At P21, 25% of the pups were expected to be GT/GT but only 14% were observed. Nevertheless, the mice alive at P21 remained alive for the rest of the study.

In our case, the survival of the GT/GT mutants is already down to just 30% at birth and none of the few GT/GT mutants born alive survive to adulthood (Rebuttal only Figure below).

There are a number of aspects that could contribute to explain the difference in survival rate. First, we obtained the SIGTR ES cell line AL0395 from the Sanger Institute and generated Ash1L GT mice independently from Brinkmeier et al. Second, the procedure to generate the Ash1L GT mice was different in the two labs. We aggregated the ES cell line with 8-cell-stage embryos of the of C57BL/6J background, then the embryos were transplanted into oviducts of recipients of the ICR background to generate chimera mice. Instead, Brinkmeier et al. injected ES cells into donor blastocysts from matings of C57BL/6J × (C57BL/6J x DBA/2J) F1 mice. Third, we performed more than 20 backcrosses in the C57BL/6J background while the available data suggest that Brinkmeier et al. have performed as low as six backcrosses. This is particularly relevant since our Ash1L GT colony showed an hypomorphic phenotype under early generation of backcrosses. Finally, the fact that the independently generated Ash1L GT mice have been maintained in different animal facilities could also have contributed to the higher lethality that we observed.

Minor comments:

- In Fig. 2, primary data from longitudinal sections would be very useful.

We included these data in the new Figure 2 of the revised manuscript as requested.

- Line 150: please clarify in the text in which setting RNA-Seq was performed, rather than referring simply to “muscle cells”.

The RNA-seq results reported in the manuscript refer to C2C12 myoblasts treated with siRNA against Ash1L compared to myoblasts treated to non-silencing siRNAs. We clarified this better in the main text as requested.

- Fig. 1: panels D and E are mislabeled in the legend.

The Figure legend has been corrected.

- For clarity, Fig. 6 would benefit from the creation of subpanels.

Done.

- Lines 138-143: panels of fig. S1 are incorrectly referred to.

This has been corrected in what is now Fig. S2.

REVIEWER#2

The Reviewer found that the *“data presented in this manuscript is very clear and interesting”*.

We thank the Reviewer for this endorsement.

However, this reviewer has some concerns with the expression of Ash1L and Ash1L KO mouse phenotypes since Ash1L KO mouse has been reported previously.

Major issues:

1. *The author mentioned that Ash1L gene KO mice generated by SIGTR ES cell line AL0395 are perinatal lethal. However, the published data utilizing the same strain of mice (Brinkmiere et al, Bio Reprod, 93(5), 121, 2015) showed normal mendelian distribution until postnatal day 7 (p7) and the number of homozygous mice declines by p21. However, around half of the homozygous mice are still alive until adult. The authors did not cite this paper. Did you see any survived Ash1L mice postnatally? If not, what is the difference between the published data and this manuscript?*

We apologize for not fully addressing this aspect.

As discussed above in answer to Reviewer#1 question 4, in our hands there is no survival of Ash1L GT mice to adulthood.

We modified the text to include the citation of the Brinkmier et al paper and provide a summary of the above explanation for the discrepancy in survival rate.

2. *It is important to show immunostaining for Ash1L during myoblast culture and muscle regeneration after CTX injection to see whether Ash1L protein is upregulated during myoblast fusion in the nucleus.*

As shown in new Figure 1D-E, we performed immunostaining showing nuclear Ash1L during myoblast culture and muscle regeneration after CTX injection.

3. *Previous work (Cole et al., Dev Cell, 7(6), 843, 2004) Cdon enhances MyoD activity through promotion of MyoD-E2A heterodimerization. Please examine whether Ash1L also promotes MyoD activity and loss of function of Ash1L reduces this activity.*

Cole et al showed that myoblasts completely lacking Cdon show a complete block in myoblast fusion and a decreased expression of the MyoD target Myogenin. In a separate set of experiments, the authors used 10T1/2 fibroblasts ectopically expressing Cdon and MyoD to show that Cdon stimulates MyoD activity by post-translationally enhancing the MyoD and E12/47 heterodimer formation. No result has been provided regarding the ability of the endogenous Cdon to stimulate MyoD activity in muscle cells.

We found that Ash1L ablation decreases Cdon expression of 50%. Accordingly, myoblasts lacking Ash1L display a significant but partial reduction in fusion. Importantly, we found no significant change in expression of the MyoD target gene Myogenin. Hence, our results suggest that Ash1L loss of function does not affect MyoD activity.

4. *Cdon was downregulated by loss of Ash1L in both primary myoblasts and C2C12 cells. Please show Cdon expression by immunostaining. In addition, please examine whether Cdon expression in skeletal muscle in both wild-type and Ash1L KO mice.*

As shown in new Figure 5, we confirmed Cdon downregulation at protein level in C2C12 cells and skeletal muscles lacking Ash1L. Moreover, the same Figure now shows that Cdon expression is positively correlated to that of Ash1L during C2C12

muscle differentiation not only at RNA but also at protein level. Unfortunately, despite testing several commercially available anti-Cdon antibodies, we failed to obtain a specific Cdon signal in human primary muscle cells.

Minor issues:

1. The authors described in the Methods section that Ash1L GeneTrap heterozygous mice, were generated by using SIGTR ES cell line AL0395, carrying an insertion of pGT01xr vector in Ash1L gene. Were these mice generated by authors using SIGTR ES cell line AL0395? Or did authors obtain mice from an organization such as Jackson Laboratories?

As discussed above in answer to Reviewer#1 question 4, we generated Ash1L GT mice independently from Brinkmier et al starting from the SIGTR ES cell line AL0395.

I hope that our modifications are considered substantially responsive to the concerns of the reviewers and I thank them once again for their attentive analysis of our work. I look forward to your advise on how to proceed.

Reviewers' Comments:

Reviewer #1:

Remarks to the Author:

This revised manuscript is further improved and adequately addresses comments by the reviewers.

Reviewer #2:

Remarks to the Author:

The revised manuscript was improved after several new experiments and rephrasing. However, the authors mentioned that "our independently generated Ash1L GT mice display perinatal lethality". However, there are still survived homozygous mice until postnatal day 8 with 20-30% ratio. Therefore, this means that perinatal lethality is not fully penetrated. The authors should mention this in the text. In addition, the table or figure showing the survival curve such as appearing in the rebuttal comments should be included in the supplemental figure.

Reviewer #1

“This revised manuscript is further improved and adequately addresses comments by the reviewers.”

We thank the Reviewer for the endorsement and for improving the quality of our manuscript.

Reviewer #2

“The revised manuscript was improved after several new experiments and rephrasing. However, the authors mentioned that “our independently generated Ash1L GT mice display perinatal lethality”. However, there are still survived homozygous mice until postnatal day 8 with 20-30% ratio. Therefore, this means that perinatal lethality is not fully penetrated. The authors should mention this in the text. In addition, the table or figure showing the survival curve such as appearing in the rebuttal comments should be included in the supplemental figure.”

As requested by the Reviewer, we modified the text to discuss the perinatal lethality of Ash1L GT mice and added the survival curve as new Supplementary Figure 2. We thank the Reviewer for the attentive analysis, which allowed to build a stronger manuscript.